# Detailed comparison of two popular variant calling packages for exome and targeted exon studies

Charles D. Warden[1], Aaron W. Adamson[2], Susan L. Neuhausen[2] and Xiwei Wu[3]

[1] Department of Computational Medicine and Bioinformatics, University of Michigan, Ann Arbor, MI, USA
[2] Department of Population Sciences, City of Hope National Medical Center, Duarte, CA, USA
[3] Integrative Genomics Core, Department of Molecular and Cellular Biology, City of Hope National Medical Center, Duarte, CA, USA

Corresponding authors
Charles D. Warden,
cdwarden@umich.edu
Xiwei Wu, xwu@coh.org

## ABSTRACT

The Genome Analysis Toolkit (GATK) is commonly used for variant calling of single nucleotide polymorphisms (SNPs) and small insertions and deletions (indels) from short-read sequencing data aligned against a reference genome. There have been a number of variant calling comparisons against GATK, but an equally comprehensive comparison for VarScan not yet been performed. More specifically, we compare (1) the effects of different pre-processing steps prior to variant calling with both GATK and VarScan, (2) VarScan variants called with increasingly conservative parameters, and (3) filtered and unfiltered GATK variant calls (for both the UnifiedGenotyper and the HaplotypeCaller). Variant calling was performed on three datasets (1 targeted exon dataset and 2 exome datasets), each with approximately a dozen subjects. In most cases, pre-processing steps (e.g., indel realignment and quality score base recalibration using GATK) had only a modest impact on the variant calls, but the importance of the pre-processing steps varied between datasets and variant callers. Based upon concordance statistics presented in this study, we recommend GATK users focus on "high-quality" GATK variants by filtering out variants flagged as low-quality. We also found that running VarScan with a conservative set of parameters (referred to as "VarScan-Cons") resulted in a reproducible list of variants, with high concordance (>97%) to high-quality variants called by the GATK UnifiedGenotyper and HaplotypeCaller. These conservative parameters result in decreased sensitivity, but the VarScan-Cons variant list could still recover 84–88% of the high-quality GATK SNPs in the exome datasets. This study also provides limited evidence that VarScan-Cons has a decreased false positive rate among novel variants (relative to high-quality GATK SNPs) and that the GATK HaplotypeCaller has an increased false positive rate for indels (relative to VarScan-Cons and high-quality GATK UnifiedGenotyper indels). More broadly, we believe the metrics used for comparison in this study can be useful in assessing the quality of variant calls in the context of a specific experimental design. As an example, a limited number of variant calling comparisons are also performed on two additional variant callers.

## INTRODUCTION

Multiple studies have previously compared variant callers for short-read sequencing data (*Bauer, 2011*; *Cheng, Teo & Ong, 2014*; *Liu et al., 2013*; *O'Rawe et al., 2013*; *Pabinger et al., 2014*; *Pirooznia et al., 2014*; *Yi et al., 2014*; *Yu & Sun, 2013*). Many studies have indicated that the variant callers available in the Genome Analysis ToolKit (GATK, *DePristo et al., 2011*; *McKenna et al., 2010*) show the best performance (*Bauer, 2011*; *Liu et al., 2013*; *Pirooznia et al., 2014*; *Yi et al., 2014*). This is in accordance with the popular use of GATK for variant calling, especially for Illumina sequencing data (*Boland et al., 2013*; *Li et al., 2014*; *Linderman et al., 2014*; *Worthey, 2013*). However, there are reports showing other variant callers that can outperform GATK, questioning the notion that GATK should be considered a gold-standard for variant calling. For example, one study found that CASAVA outperformed the GATK UnifiedGenotyper when calling single-nucleotide variants (*Cheng, Teo & Ong, 2014*) and another study showed that a novel algorithm called Scalpel outperformed GATK HaplotypeCaller for indels (*Narzisi et al., 2014*). The use of multiple-variant callers has also been proposed (*Lam et al., 2012*; *O'Rawe et al., 2013*; *Pabinger et al., 2014*; *Yu & Sun, 2013*), but this will increase the run-time (or at least computational resources) necessary for analysis (which can be especially important for large patient cohorts).

While the somatic variant calling function in VarScan (*Koboldt et al., 2009*; *Koboldt et al., 2012*) has been compared against other somatic variant callers (*Roberts et al., 2013*; *Wang et al., 2013*; *Xu et al., 2014*), most comparisons for single-sample variant calling (often used for identifying germline mutations) did not directly compare GATK variant callers against VarScan. One study indicated that VarScan was less accurate than the other variant callers (*Cheng, Teo & Ong, 2014*). Another study showed VarScan as being more similar to the other variant callers but still ranked GATK as the best option (*Yi et al., 2014*). However, VarScan has been used for variant calling in a large number of studies (*Worthey, 2013*), and we hypothesize that the simple, intuitive parameters can be helpful in establishing an optimal set of variants for a given dataset. Also, as emphasized in this study, the run-time for VarScan should typically be shorter than GATK. Therefore, we wished to determine if (1) the previous VarScan benchmarks can be reproduced in our own analysis and (2) if use of non-standard parameters can improve the quality of the VarScan variant calls.

This study also examines the relative impact of pre-processing steps in GATK (specifically, the indel realignment and quality score base recalibration steps). Therefore, we compared variant calls with GATK and VarScan for each step separately, with both pre-processing steps, as well as without either pre-processing step. There has been at least one previous study to showing that filtering can improve the quality of variant calls (*Carson et al., 2014*), beyond the GATK quality score base recalibration. We assessed whether running VarScan with different sets of parameters (using three different parameters settings: see Methods) can also increase the accuracy of the resulting variant calls. Additionally, we have used a simple filtering strategy for GATK variants (looking at all variants called versus filtering out variants with a low quality flag), so there are two sets of variant lists for both GATK UnifiedGenotyper and GATK HaplotypeCaller.

In order to avoid bias that could come from studying only a limited number of samples, variant calls were performed on 14 targeted exon (for 1000 genes) samples from *The 1000 Genomes Project Consortium (2012)*, 12 exome samples from *The 1000 Genomes Project Consortium (2012)*, and 15 Illumina exome samples from SRP019719 (*O'Rawe et al., 2013*). We believe this helps yield robust results both in terms of the number of samples studied per cohort as well as variations in study design (i.e., the method of targeted sequencing). The 1000 Genomes study was specifically chosen in order to test recovery of validated variants as well as to compare concordance between samples subject to both targeted exon and exome sequencing.

In short, this study presents a detailed characterization between GATK and VarScan on 41 samples (with varying target designs), where each sample has 28 variant lists for comparison. Variant lists are compared based upon the number of variants called, the proportion of novel variants (defined in those absent from *The 1000 Genomes Project Consortium (2012)*, Exome Sequencing Project (*Fu et al., 2013*), and dbSNP (*Sherry et al., 2001*)) in the variant list, and the reproducibility of variant calls using different technologies. A limited number of additional comparisons are also performed in order to help illustrate how these metrics can be used to select the optimal variant caller for a given dataset. This analysis demonstrates that a conservative set of parameters (referred to as "VarScan-Cons") can be used to produce a reproducible list of variants from VarScan, and there is limited evidence that VarScan-Cons has a lower false discovery rate among novel variants. This study also presents evidence that the GATK HaplotypeCaller may have a higher false discovery rate in calling indels compared to the GATK UnifiedGenotyper.

## MATERIAL AND METHODS

### Sample selection

All datasets were downloaded as .fastq files from the European Nucleotide Archive (*Leinonen et al., 2011*). Illumina exome samples were downloaded from SRP019719 (*O'Rawe et al., 2013*). *The 1000 Genomes Project Consortium (2012)* data, abbreviated as 1KG in this manuscript, was selected on the basis of having (1) exome data, (2) targeted exome data, (3) Omni SNP chip data, and (4) validated SNPs. Among samples meeting those criteria, 12 samples were selected based upon (1) their presence in disparate populations (CEU: Northern and Western European Ancestry, CHB: Han Chinese, JPT: Japanese, and YRI: Yoruba/African) and (2) maximum number of validated SNPs within each of the four selected populations.

1000 Genomes validated SNPs and Omni SNP chip .idat files were downloaded from the 1000 Genomes FTP site (ftp://ftp.1000genomes.ebi.ac.uk/vol1/ftp/). Likewise, target design files (for targeted exon and exome samples) were downloaded from this site. More specifically, the phase 3 design files were used to calculate coverage statistics for the exome samples, and validated variants come from ALL.chr20.exome_consensus_validation_454.20120118.snp.exome.sites.vcf.gz (pooled 454 PCR sequencing data). At the time this dataset was downloaded, the hg19 reference location for validated SNPs was off-set by one (similar to indels in .vcf files), and this was taken into consideration

during analysis. Targeted gene coordinates (for hg18) from P3_consensus_exonic_targets.bed were converted to hg19 coordinates using the LiftOver function in Galaxy (*Blankenberg et al., 2001*; *Giardine et al., 2005*; *Goecks et al., 2010*).

## Data processing

Reads were aligned to a karotype-sorted hg19 reference (necessary for running GATK (*McKenna et al., 2010*)) using BWA (v0.7.5a) (*Li & Durbin, 2009*). Prior to variant calling, singletons were filtered out using samtools (v0.1.19) (*Li et al., 2009*), .bam files were coordinate sorted via Picard (v1.105, http://picard.sourceforge.net/), and duplicates were removed via Picard. Prior to running GATK, read groups were added via Picard, and .bam file was re-ordered according to chromosome karyotype. Prior to running VarScan (*Koboldt et al., 2012*), a pileup file was created via mpileup in samtools (*Li et al., 2009*) and positions without any aligned reads were filtered out. These represent the minimum pre-processing sets and are labelled as "No Preprocessing". Alignment statistics for these samples are shown in supplementary tables (Table S1 for 1KG Targeted Exon, Table S2 for 1KG Exome, and Table S3 for SRP019719).

There are three additional pre-processing pipelines that were considered for VarScan and GATK comparisons. "Realign Only" runs an indel realignment using GATK (using the RealignerTargetCreater and IndelRealigner functions). "Recalibrate Only" uses GATK to recalibrate quality scores (using the BaseRecalibrator and PrintReads functions). "Full Pipeline" runs the indel realignment functions and then performs base recalibration.

MiSeq amplicon data (for individual K8108-49685s in the SRP019719 dataset) was processed in a similar pipeline as the array-based targeted sequencing data except that reads were first trimmed to 150 bp and PCR duplicates were not removed (since all reads were PCR duplicates). Also, unlike the SRP019719 exome data, GATK did not detect miscoded quality scores in the SRP019719 amplicon data (so, the variant calling steps match the 1000 Genomes sample commands in Text S1).

Human610 SNP chip data for individual K8101-49685s (paired with SRP019719 Illumina exome sample SRX265476) was reported in the Illumina "TOP" format, so the reverse complement of the allele was used when the IlmnStrand and RefStrand did not match (as defined in the human610-quadv1_h.csv manifest file). Allele sequences were provided without respect to a reference sequence, so SeattleSeq Variant Annotation (http://snp.gs.washington.edu/SeattleSeqAnnotation138/) was used to determine the reference sequence to focus on variants that differ from the reference sequence (to make results comparable to the Illumina sequencing variant calls). This is the latest version of the manifest file, but it was designed using dbSNP 131 and some discordant SNPs were due to annotations where the forward strand may vary from the hg19 reference sequence. However, this only affected a relatively small minority of SNPs (<5% of variants).

Raw .idat files for 1000 Genomes Omni SNP chip samples were processed in Illumina® Genome Studio™(V2011.1). A pre-defined clustering file (HumanOmni2.5-4v1-Multi_H.egt) was used to call genotypes. Variants were exported in the "Plus" format (so, no genotype conversion was necessary). Samples were annotated using the

HumanOmni2.5-4v1-Multi_B.bpm manifest file, including the genomic position in hg18 coordinates. Coordinates were converted to hg19 via liftOver in Galaxy (*Blankenberg et al., 2001*; *Giardine et al., 2005*; *Goecks et al., 2010*) and reference sequences were determined for all on-target probes ($n = 2,257$) via SeattleSeq Variant Annotation (http://snp.gs. washington.edu/SeattleSeqAnnotation138/). Each sample had 2,179–2,188 genotyped SNPs recognized by SeattleSeq, with 477–616 non-reference alleles per sample.

## Concordance definitions

Concordance was defined as recovery of a set of lower-throughput variants (validated, SNP chip, targeted exon, amplicon), except for the targeted exon technical replicates where concordance was defined as (2 ∗ Number of Overlapping Variants)/(Sum of Variants from both samples). For clearity, we therefore refer to recovery of lower-throughput results as the "recovery rate" in the results and reserve the term "concordance rate" for only the technical replicate analysis. When defining variant concordance between the SNP chip data and exome data, recovery of the known variant is counted as a concordant variant (even if multiple variants are called at a given position).

Unlike most comparisons in this study, the SNP chip comparisons do not specifically focus on the coding variants. As such, extraction of SNP chip variants within the target regions currently includes some non-coding variants (such as intronic variants) that would falsely be called discordant if focusing only on coding variants. SNP chip variants have been filtered to only include variants that vary from the reference sequence.

## Calling variants

Variants were called using GATK (v.2.8.1) based upon established best practices (*DePristo et al., 2011*; *Van der Auwera et al., 2002*). Variants were called using both the UnifiedGenotyper (UG) and the HaplotypeCaller (HC). For variant characterization, the set of all variants was considered (labelled as UG-all in figures for the UnifiedGenotyper and HC-all for the HaplotypeCaller) as well as only the high-quality variants that didn't contain the "LowQual" flag in the .vcf file (labelled as UG-HQ in figures for the UnifiedGenotyper and HC-HQ for figures for the HaplotypeCaller). In addition to the parameters described in the GATK best practices, variant calling for SRP019719 also required some additional parameters due to the quality scores for the ENA reads (these extra parameters were not necessary for calling variants from 1000 Genomes Project data). Parameters for calling 1000 Genomes and SRP019719 variants are provided in Texts S1 and S2, respectively.

VarScan (v.2.2.8) variants were called using pileup2snp and pileup2indel. Three different sets of parameters were used for calling variants. "VarScan: Default" specifies no additional parameters beyond the minimal requirements. "VarScan: *p*-value" sets a *p*-value threshold of 0.05, but specifies no additional parameters. "VarScan: Conservative" uses the following parameters to stringently call variants: minimum 10 total reads at the position of interest, minimum of 4 supporting reads to call variant, minimum average quality of 20, and minimum variant allele frequency of 0.3. Please see Table 1 for a summary of parameters used for the VarScan comparisons. A template for running VarScan-Cons

**Table 1 Parameter settings for VarScan comparisons.** Unless a *p*-value cutoff is specified, VarScan doesn't calculate *p*-values.

|  | VarScan-Default | VarScan-*p*-value | VarScan-Cons |
|---|---|---|---|
| Minimum coverage | 8 | 8 | 10 |
| Minimum supporting reads | 2 | 2 | 4 |
| Minimum average quality | 15 | 15 | 20 |
| Minimum variant frequency | 0.01 | 0.01 | 0.3 |
| Maximum *p*-value | (none) | 0.05 | (none) |

(Exon_Capture_workflow_v2.pl) is provided at https://sites.google.com/site/cwarden45/scripts.

Additional variant callers were also applied to the "No Preprocessing" alignments and the "Full Pipeline" alignments. Freebayes (*Garrison & Marth, 2012*) was applied to the 1000 Genomes Exon Targeted samples, using default parameters. The Bayesian variant caller in the bcftools function (in the samtools package (*Li et al., 2009*)) was applied to all three datasets, using default parameters (followed by applying vcfutils.pl with a maximum read depth of 200). Unlike GATK and VarScan, samtools has a unique indel format to represent ambiguous indels. Although the ANNOVAR file conversion program can remove all nucleotides that are not part of the indel, the genomic position used to represent this indel is not necessarily the same as GATK and VarScan. For this reason, we only present the samtools SNP results in this manuscript. All analysis was performed on a shared Linux server with concurrent usage ($\times$64, CentOS 5.10, 256 GB RAM, 4 CPU, 8 cores each, $6 \times 2.27$ GHz processors).

Because a publication on the 1000 Genomes exome and targeted exon datasets has not yet been published, we are only reporting variant frequencies for a single chromosome (chr20) in order to comply with 1KG publication requirements. Thus, only 1,140,996 base pairs of targeted sequence is considered for variant call benchmarks in the 1000 Genomes exome datasets, and only 35,309 base pairs are considered for variant call benchmarks in the 1000 Genomes exon targeted samples. In contrast, the SRP019719 comparisons are genome-wide (with a targeted design covering 46,401,093 base pairs).

## Annotating variants

After variants were called, ANNOVAR (*Wang, Li & Hakonarson, 2010*) was used to determine the population frequency for each variant, using the summarize_annovar.pl function that was last updated 2/11/2013. Variants were defined as "low frequency" if *The 1000 Genomes Project Consortium (2012)* frequency and NHLBI Exome Sequencing Project (*Fu et al., 2013*) frequencies were both less than 0.01. Variants were defined as "novel" if they were not present in any 1KG or ESP samples as well as undefined in dbSNP (*Sherry et al., 2001*). Some rare variants may not truly be truly novel, but this distinction between "novel" and "previously observed" variants should typically be valid. Variants were predicted as damaging if the SIFT (*Ng & Henikoff, 2003*) score greater than or equal to 0.95 or the PolyPhen (*Adzhubei et al., 2010*) score was greater than or equal to 0.85

(which are the thresholds used by ANNOVAR to flag a variant as damaging, for those two programs). Although population frequency and damaging frequencies can be provided for both SNPs and indels, there were very few indels within on-target regions for the targeted exon samples (so, ANNOVAR characterization is only present for SNPs). ANNOVAR also provides a common format that can compare all variant callers (and it allows easy determination of variants included within the targeted exon panel), so all between-sample comparisons were performed using the ANNOVAR exome summary table (except for the SNP chip comparison, which uses the genome summary table).

## RESULTS

### Replication of GATK and VarScan variant calls among technical replicates

There were two 1000 Genomes (1KG) individuals that had two targeted exon samples (NA18637: SRR013654 + SRR013709, and NA18510: SRR017908 + SRR018122). We treated these samples as technical replicates in order to assess the reproducibility of variant calls from GATK and VarScan. All samples contained Illumina GAII sequencing reads from the Broad Institute (Table S1). The SNP concordance between samples was clearly higher for NA18637 than NA18510 (Fig. S1A). These trends hold true when only coding variants were considered, either for all genes (Fig. S1B) or only coding variants within targeted genes (Fig. S1C). In fact, the concordance rate increases when focusing on coding variants. Similar statistics were provided for indels (Fig. S2), but the sample size was too small to compare indels with more focused variant sets. It is unlikely that one NA18510 sample was simply mislabelled: the concordance between exome and targeted exon samples was more similar (Tables S4 and S5), and we would expect lower concordance between two random individuals. One possible explanation is that sample SRR017908 had 36 base pair reads (instead of 76 base pair reads), which could explain the lower concordance among the NA18510 samples (which have varying read lengths, unlike the NA18637 samples). Also, it is worth noting that the concordance of indel calls was better for the GATK UnifiedGenotyper (and usually VarScan) than the GATK HaplotypeCaller (Fig. S2), and this was true for both samples. This may corroborate the conclusions of a previous study indicating an increased false discovery rate causes the GATK HaplotypeCaller to produce a larger number of novel variants (*Lescai et al., 2014*).

### GATK and VarScan show similar reproducibility among validated and SNP chip variants

The recovery rate for validated SNPs in 1KG samples was similar for each variant calling pipeline (Fig. 1, Table S6). Statistics are only provided for the 1KG exome samples, because no validated SNPs occur in the targeted regions for the targeted exon samples (for the samples selected for this study). Out of the total 35 validated SNPs present among the 12 exome samples, recovery rates varied between 80–94%. VarScan-Default had the highest overall sensitivity and VarScan-Cons had the lowest overall sensitivity. The pre-processing steps had little impact on the results: in fact, the only effect was that base recalibration

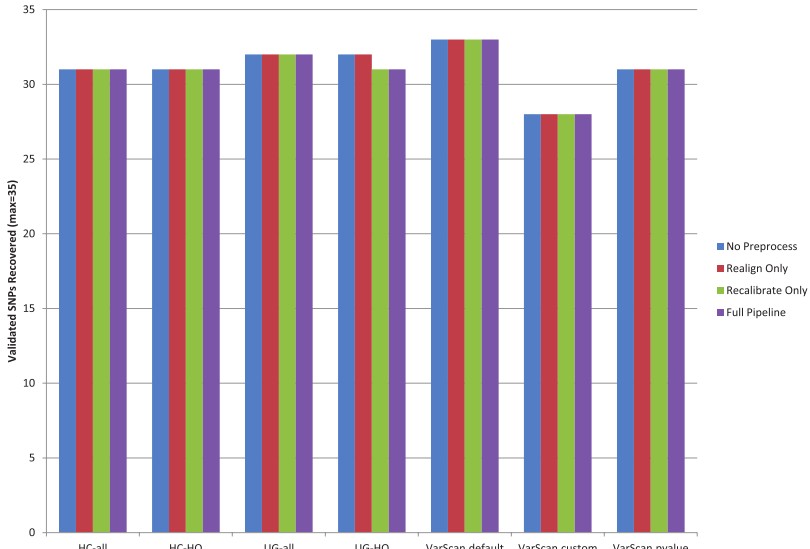

**Figure 1 Recovery of 1KG 454 PCR validated SNPs (chr20).** A pooled set of 35 validated variant from the 1000 Genomes exome sample characterized in this study ($n = 12$) was used to assess the sensitivity of various variant calling algorithms. The 1000 Genomes targeted exon samples were not compared because no validated variants were covered in targeted regions for that design. Seven variant calling strategies were tested (GATK UnifiedGenotyper and HaplotypeCaller, with and without filtering low quality variants; VarScan with 3 sets of parameters, see Methods). "VarScan-Cons" is the most conservative set of parameters for VarScan. Each variant caller was also tested with 4 preprocessing conditions: variants called using both GATK indel realignment and quality score recalibration ("Full Pipeline"—purple), indel realignment only ("Realign Only"—red), quality score recalibration only ("Recalibrate Only"—green), or neither ("No Preprocess"—blue). Publically available validated variants are only available for chr20, so this is the maximum number of validated SNPs that can be characterized for these samples. The validation status for each individual SNP under each variant calling condition is provided in Table S7. Validated variants were never called for chr20:3193991 for individual NA18505 (exome sample SRX237141, covered $81\times$ with the reference allele in all reads) or for chr20:57769739 for individual NA18532 (exome sample ERR031956, not covered by any reads but located in the coding sequence for ZNF831).

caused one fewer validated SNP to be recovered when using the GATK UnifiedGenotyper. Unfortunately, this is a limited number of validated SNPs, so it is difficult to say how closely these are tied to the true sensitivity rates.

SNP chips can also be used to assess sensitivity for variant lists. However, it should be noted that these will mostly be common SNPs, which importantly means that recovery of SNP array variants may not represent the accuracy of rare variant calls. Nevertheless, it is useful to see how the SNP chip recovery compared to the validated SNPs and the total SNP calls. In most cases, all variant calling algorithms could recover >85% of Omni SNP chip variants from the 1000 Genomes exome samples (Fig. 2A). Similarly, we compared variant calls for a subject from the SRP019719 dataset that had both Illumina exome and SNP chip data: again, the majority (88–96%) of SNP chip variants were recovered in the paired exome dataset (Fig. 2B, Table S7). In both cases, recovery was restricted to SNP chip variants within targeted regions for the exome sample. It is also worth noting that SNP chip alleles matching the reference sequence are not considered, and the recovery rate would of course be higher if these sites were considered. Unlike the validated variants (Fig. 1),

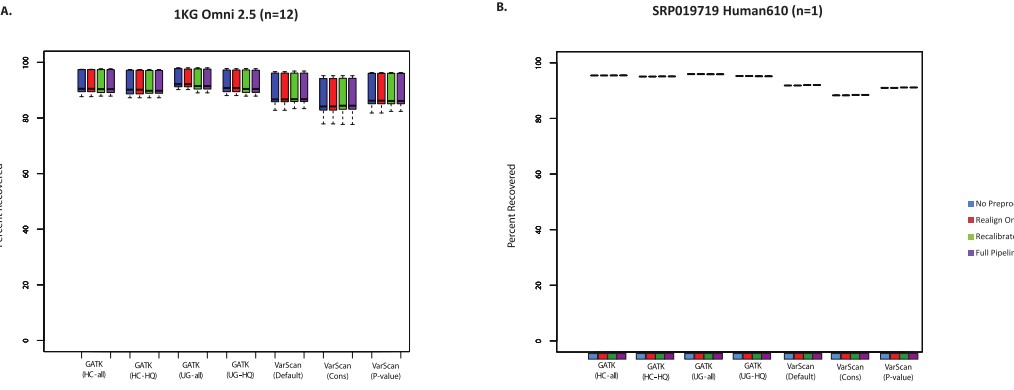

**Figure 2** (A) Recovery of variants reported from the Omni SNP chip for 1000 Genomes exome samples ($n = 12$). Only alleles that varied from the reference sequence and were located within the regions targeted in the exome sequencing design were considered for this analysis (this varied somewhat between samples, between a minimum of 477 variants and a maximum of 616 variants). Seven variant calling strategies were tested (GATK UnifiedGenotyper and HaplotypeCaller, with and without filtering low quality variants; VarScan with 3 sets of parameters, see Methods). "VarScan–Cons" is the most conservative set of parameters for VarScan. Each variant caller was also tested with 4 preprocessing conditions: variants called using both GATK indel realignment and quality score recalibration ("Full Pipeline"—purple), indel realignment only ("Realign Only"—red), quality score recalibration only ("Recalibrate Only"—green), or neither ("No Preprocess"—blue). (B) Same as A, but for 6437 variants on a different SNP chip design (Human 610) compared to exome variant calls for sample SRX265476.

VarScan (using default parameters) no longer had the highest sensitivity; instead, the average recovery of SNP chip variants was slightly higher for GATK. In short, we believe that all variant calling strategies show a similar recovery rate for validated SNPs and SNP chip variants, with a false negative rate for these common variants likely being less than 15%.

## False positives are enriched for novel variants

The recovery of targeted exon variants (within the set of targeted genes) among the exome samples was also typically quite high (Fig. 3A). The main exception was for variants called using VarScan with default parameters, and this was also true for the VarScan calls with the *p*-value filter (to a lesser extent). In this case, it is important to note that the targeted exon calls are not truly a gold standard. In other words, the lower recovery rate can be due to a high false discovery rate among the targeted exon variant calls. For example, position 17933286 on chromosome 20 for individual NA18566 was covered by 189 reads in the targeted exon sample and 109 reads in the exome sample: 90% of reads match the reference sequence in the targeted exon sample and 99% of the reads match the reference sequence in the exome sample. Using the default parameters, VarScan calls a variant "G" allele in the targeted exon sample (which is present in 12/189 reads). However, this is likely a normal diploid individual with the true genotype of T/T at this position, with deviations from the reference sequence that are probably due to technical error (where the proportion of errors found at a particular site can randomly fluctuate between samples). While this specific variant may serve a useful conceptual example, it is important to identify subsets of variants are likely to drive the lower recovery rate among the VarScan-Default variant calls.

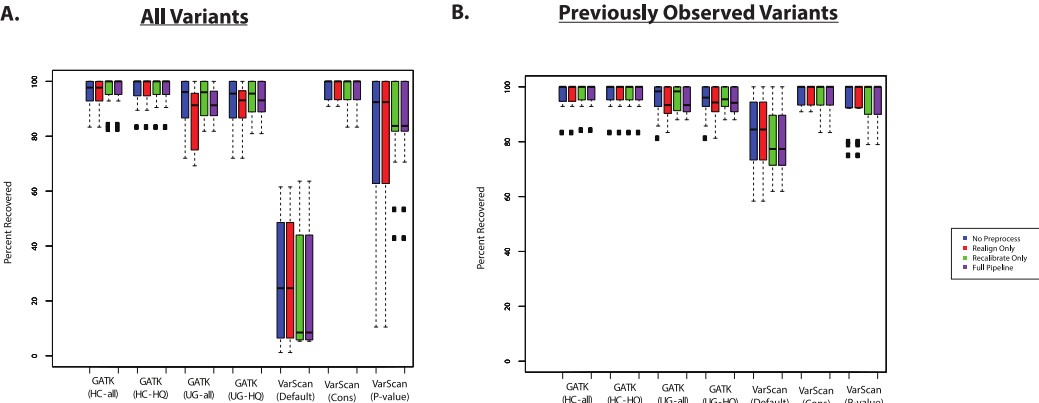

**Figure 3** **Recovery of coding SNPs from targeted exon samples (1KG Exome—chr20).** (A) SNP calls from paired targeted exon and exome datasets were compared to test the robustness of the calls made in the targeted exon data. Indel calls are not presented because there are almost no coding indels on chr20 for the targeted exon datasets. Two subjects have two targeted exon datasets (Tables S1 and S2), and concordance with exome datasets was reported for both targeted exon datasets separately (resulting in 14 concordance values per variant calling strategy). Seven variant calling strategies were tested (GATK UnifiedGenotyper and HaplotypeCaller, with and without filtering low quality variants; VarScan with 3 sets of parameters, see Methods). "VarScan-Cons" is the most conservative set of parameters for VarScan. Each variant caller was also tested with 4 preprocessing conditions: variants called using both GATK indel realignment and quality score recalibration ("Full Pipeline"—purple), indel realignment only ("Realign Only"—red), quality score recalibration only ("Recalibrate Only"—green), or neither ("No Preprocess"—blue). Concordance is reported as recovery of SNPs called in the targeted exon data, but these cannot be treated as "gold standard" variant calls. Most clearly, there was a high false positive rate when running VarScan with default parameters, so a high proportion of those variants called in the targeted exon samples could not be recovered in the exome dataset. In fact, on-target coverage is typically lower for the targeted exon samples than the exome staples (Tables S1 and S2). (B) Same as (A), but only previously observed variants are included in the percent recovery calculation.

If variant lists are confined to previously observed variants (present in the 1000 Genomes Project, Exome Sequencing Project, or dbSNP), then recovery of targeted exon VarScan-Default SNPs are considerably improved. For example, VarScan calls with the $p$-value filter were now comparable to the other variant callers (Fig. 3B). An increased false discovery rate has been observed for loss of function variations due to the presence of uniform sequencing errors and non-random distribution of natural variation (*MacArthur et al., 2012*), and we believe a similar concept applies more broadly to novel variants. Accordingly, the recovery rate was extremely low for novel variants called using VarScan with default parameters (Table S8). There were no novel VarScan-Cons variants called within the coding regions of chromosome 20, but it was likely difficult to call true novel variants given that variants called by the 1000 Genomes Project are defined as previously observed in our analysis. The recovery rate was also relatively low for other variant callers, but there are only a small number of novel variants occurring within targeted regions of chr20 for most of the lists of variant calls.

One sample in the SRP019719 dataset underwent exome sequencing as well as amplicon sequencing for a selected number of variants. Although limited to a single sample, this comparison was important because the amplicon dataset contained more novel

variants than present within 1000 Genomes targeted exon genes on chromosome 20. The amplicon comparison showed a universal decrease in the recovery of novel variants (Fig. S3), although it should be noted that the overall recovery rate was also lower than the array-based enrichment samples. This is most likely because duplicate reads were not removed, and errors introduced during PCR amplification increased the total number of false positives in the amplicon dataset. For example, the targeted variants were located towards the middle of each amplicon, but sequencing errors may be more frequent towards the end of the amplicons. This expectation of an increased false discovery rate is supported by the fact that the number of SNPs called in the amplicon dataset (Table S9) is much higher than the number of variants that were targeted for validation (O'Rawe et al., 2013). Accordingly, the novel variant frequency is much higher for the amplicon datasets than the exome and targeted exon datasets (Table S10). Of course, changing variant calling criteria (such as imposing stricter coverage requirements) will affect the number of false positives (and thus the novel variant rate), so this result should not be interpreted as an indication that it is impossible to get high quality results from the amplicon dataset. Instead, this result emphasizes that benchmarks will vary with different target designs, while still demonstrating the general increase in false discovery rate for novel variants (although the extent of enrichment can vary with target design and variant caller).

In the example of the specific variant at position 17933286 on chromosome 20, there was a very small proportion of reads containing the variant allele. If the density distributions of novel and previously-observed variations are compared, it is clear that most of the novel variants called by VarScan-Default had less than 20% of reads containing the variant allele (Fig. 4 and Fig. S4). We expect the most conservative aspect of VarScan-Cons is the requirement that variants be present in at least 30% of reads. To test this hypothesis, we calculated the recovery rate for targeted exon VarScan-Default variants with varying thresholds for the percentage of reads containing the variant allele as well as the minimum number of reads with the variant allele. There were no novel variants called by VarScan-Cons and all novel variants supported by less than 30% of reads had significantly lower recovery than previously-observed variants with comparable support (Table S11, Fig. S5). Recovery rates among previously observed variants dropped when a threshold of 50% supporting reads was used because this threshold splits the heterozygous peak in half (increasing the likelihood of encountering variants that marginally meet the criteria in one dataset but marginally do not meet the criteria in the other dataset). In short, novel variants often have an increased false discovery rate because the low likelihood of sequencing error occurring for the same nucleotide at the same position of a naturally occurring variation, where most novel variants (defined using sufficiently liberal criteria) fit the model of a sequencing error that will be observed most frequently in variants with small percentages of supporting reads.

The difference in density distributions was most clear for variant with less than 20% supporting reads, but the entire peak of variants with less than 30% supporting reads violates an assumption a diploid human genome. In other words, reads matching the reference genome were not called as variants, so the homozygous wild-type allele should

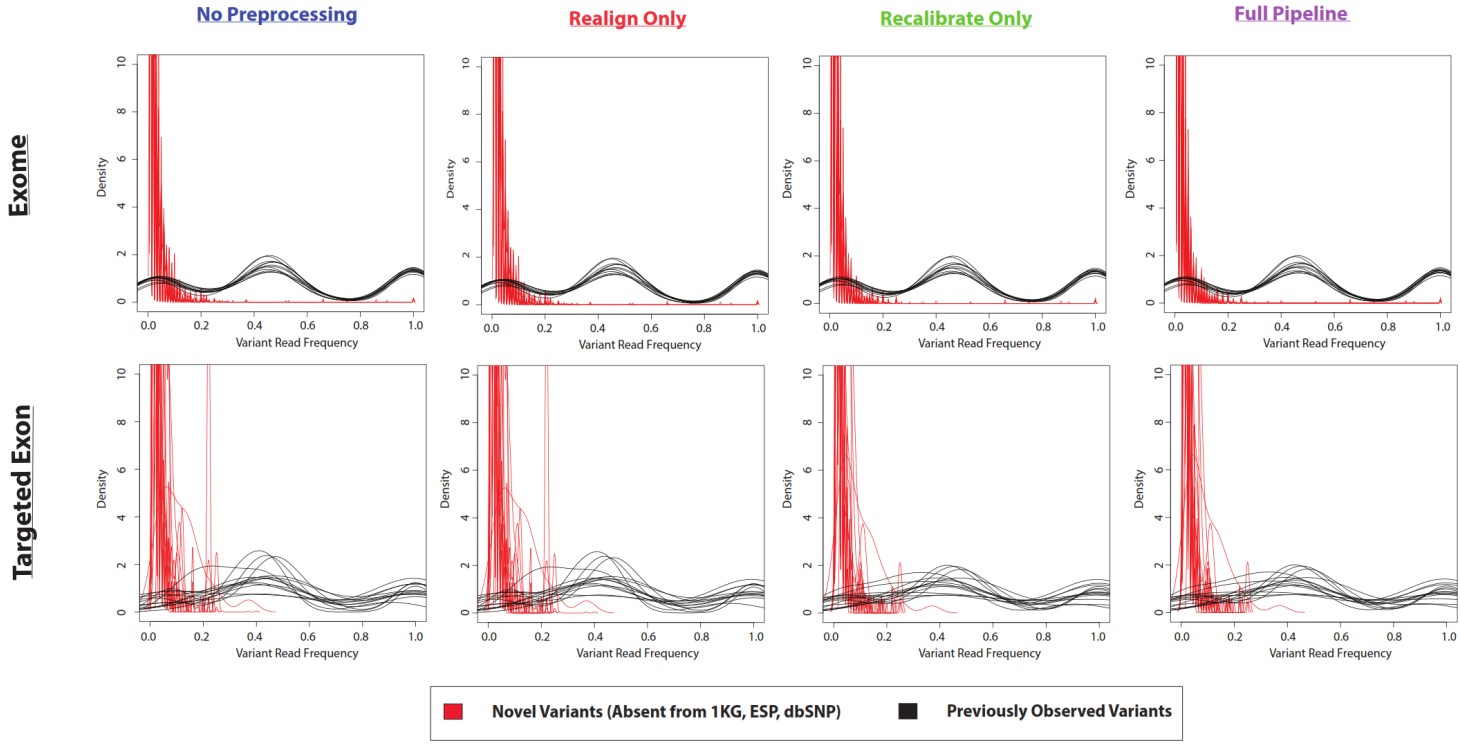

**Figure 4 Frequency of reads supporting coding VarScan-Default SNP variants (1KG Exome—chr20).** Density plots for novel variants (those variants not reported in the 1000 Genomes Project, the Exome Sequencing Project, or dbSNP) are shown with red lines while density plots for previously observed variants are shown with red lines. Notice that most novel variants have a very low frequency of supporting reads, with a trend so strong that it cannot be observed on the scale of the current figure (although a rescaled figure is shown in Fig. S4, to emphasize the radically different density distributions). We believe that this because the majority of variants with less than 30% supporting reads are due to sequencing errors, which is why the height of the peak (in Fig. S4) approximately matches the error rate (∼1%). This distribution looks similar regardless of pre-processing pipeline or target design (e.g., exome versus targeted exon).

not have a peak in the density distribution. This is the basis of selecting this threshold for VarScan-Cons. As expected, there were only two peaks for the density distribution of supporting reads for VarScan-Cons (Fig. 5). This was seen most clearly for the SRP019719 exome data, where there were a sufficiently large number of novel variants to define clean density distributions. Importantly, this was also true after removing low-quality GATK UnifiedGenotyper and HaplotypeCaller (Fig. 5), which indicates that most variants with less than 30% supporting reads were also removed using this independent variant calling strategy. This particular supporting read threshold was specifically designed with a diploid organism in mind, but technical errors should always be enriched among all variants with low percentages of supporting reads. Therefore, visualization of density distributions for supporting read frequencies of novel variants can be a useful strategy for comparing variants lists for any organism.

## Estimation of accuracy for GATK and VarScan variant calls

Although 1KG samples were selected on the basis of having some validated positive controls, it is useful to have quality control metrics that will be correlated with the true sensitivity and specificity for a given variant caller. Each strategy has notable caveats for

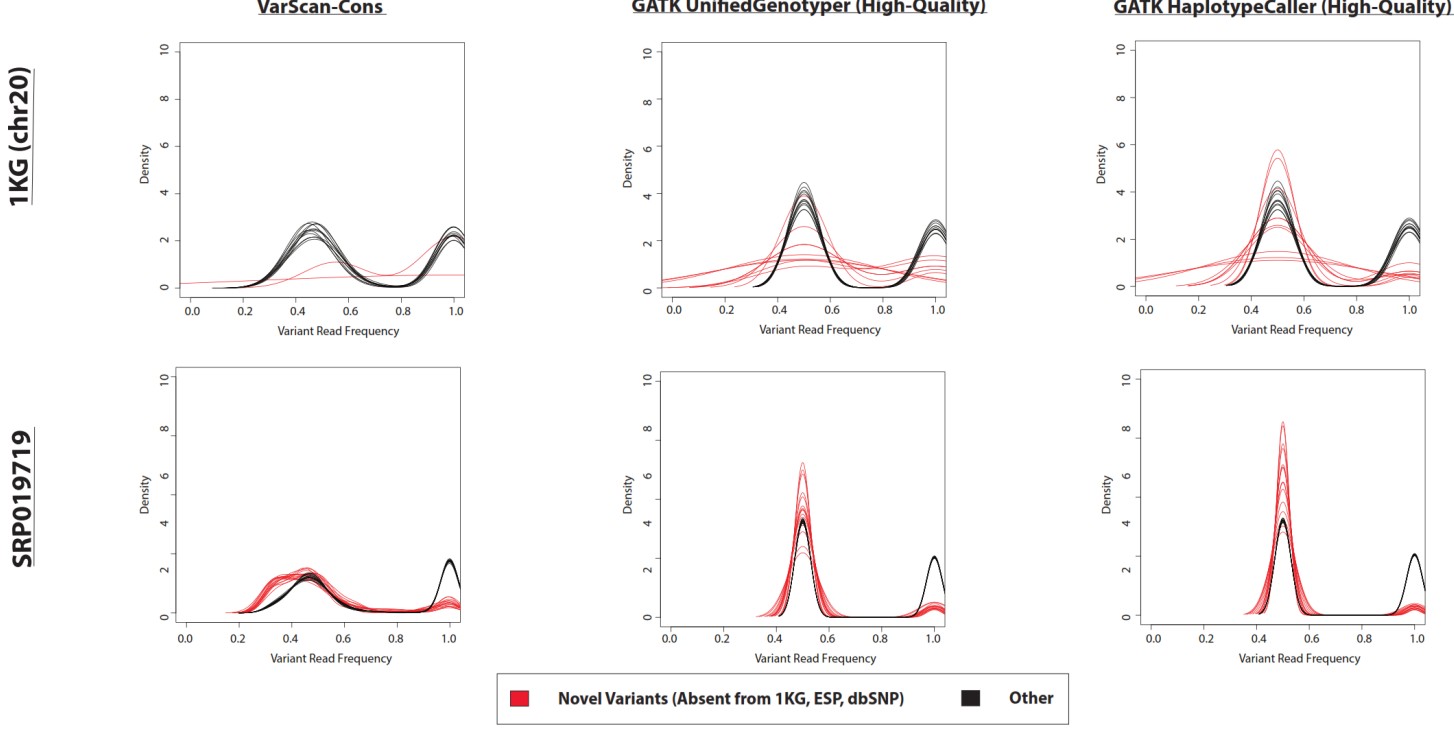

**Figure 5 Frequency of reads supporting SNP variants in exome datasets.** Density plots for novel variants (those variants not reported in the 1000 Genomes Project, the Exome Sequencing Project, or dbSNP) are shown with red lines while density plots for previously observed variants are shown with red lines. Density plots are shown for two datasets (1000 Genomes and SRP019719) with 3 lists of variants (VarScan-Cons, high-quality GATK UnifiedGenotyper, and high-quality GATK HaplotypeCaller). Density plots are only created for variant lists with greater than 2 variants, so some samples do not have novel variant density plots. Notice that the high-quality GATK variants lists lack variants with low percentages of variant reads (as was the case for VarScan-Default, Fig. 4), similar to VarScan-Cons. This shift in supporting read frequencies correlates with the concordance rates (Fig. 3) and the novel variant frequencies (Fig. 7) for the corresponding variant lists.

interpretation, but we think it is useful to have quality control metrics that can be used to select an optimal variant caller for a given dataset.

First, we assume that the total number of variants is correlated with sensitivity. Of course, the accuracy of this assumption would depend upon the false discovery rate for the variant caller. Nevertheless, we believe that the total number of called variants is a useful benchmark to compare variant callers. SNP and indel counts are shown for 1KG exome and targeted exon samples in Fig. 6. In most cases, the pre-processing pipeline had a minimal effect on the size of the resulting list of SNPs, with the notable exception of the GATK UnifiedGenotyper (although the impact was significantly decreased for high-quality SNPs). Similarly, the overlap was strong for variants called without these extra pre-processing steps (Figs. S6 and S7). However, it is difficult to tell how common this trend is for all datasets: for example, the exome samples from SRP019719 did not show this same difference (Fig. S8) and instead showed a relatively greater difference in the number of SNPs called by VarScan with default parameters. The size of the SNP lists clearly varied with different VarScan parameters: the *p*-value threshold considerably decreased the number of SNPs called, the conservative parameter set was even more restrictive, and

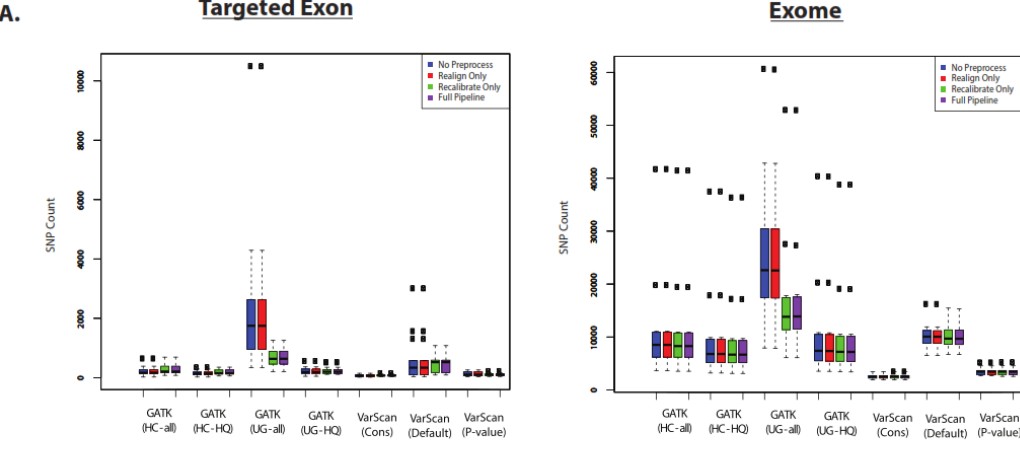

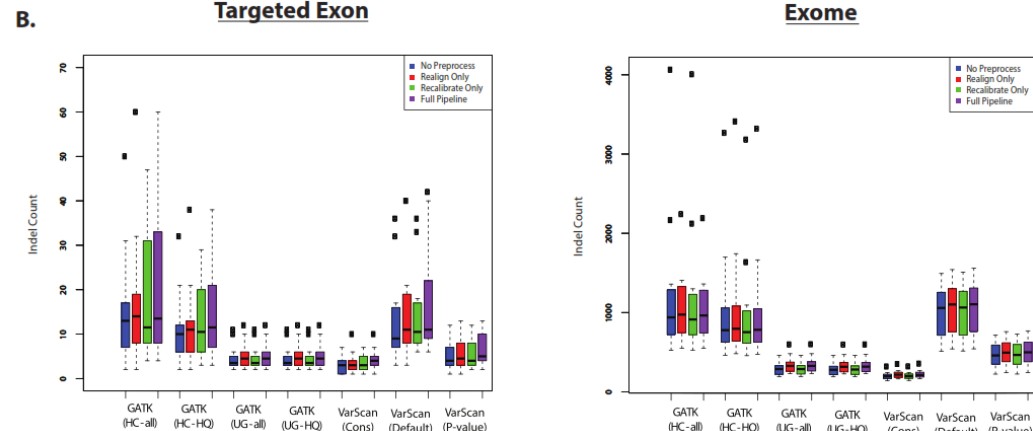

**Figure 6 Number of variants called for GATK versus VarScan (1KG—chr20).** (A) Number of SNPs called for selected 1000 Genomes (1KG) targeted exon samples ($n = 14$) and exome samples ($n = 12$). The number of SNPs called is displayed for variants called using both GATK indel realignment and quality score recalibration ("Full Pipeline"—purple), indel realignment only ("Realign Only"—red), quality score recalibration only ("Recalibrate Only"—green), or neither ("No Preprocess"—blue). Variant counts are provided for the GATK UnifiedGenotyper ("UG"), GATK HaplotypeCaller ("HC"), and VarScan using 3 sets of parameters (see Methods). UnifiedGenotyper and HaplotypeCaller variants are then divided into the set of all variants ("UG-all" and "HC-all") and higher-quality variant calls where variants flagged as low quality have been removed ("UG-HQ" and "HC-HQ"). "VarScan-Cons" is the most conservative set of parameters for VarScan. These values represent the total number of SNPs called. (B) Same as (A), for indels instead of SNPs.

both results run with non-default settings produced significantly fewer SNPs than GATK. Among the high-quality calls, the number of SNPs was similar for GATK HaplotypeCaller versus GATK UnifiedGenotyper. The relative number of indels varied from the relative number of SNP calls: however, there were only a limited number of indel calls on chr20 for the targeted exon panel, so we think the number of 1KG exome indel calls were more reliable for assessing general trends. There was a modest but noticeable increase in the

Peer

number of indels called if the indel realignment was run, and the GATK HaplotypeCaller produced a much larger number of indels than the GATK UnifiedGenotyper. Similar to the SNP calls, VarScan produces fewer indel calls when using more stringent parameters.

Second, we assume that most variants should have been previously observed in large scale sequencing project (like the 1000 Genomes Project and the Exome Sequencing Project) and variant databases (like dbSNP), and that an over-representation of variants of unknown frequencies (e.g., novel variants) should correspond to an increased false discovery rate. Of course, some novel variants will in fact be accurate and the proportion of true novel variants will vary with demographics (subjects from ethnicities that have been characterized in greater detail will likely show fewer novel variants, and vice versa). However, some proportion of novel variants are clearly unacceptable: for example, more than 50% of the variants called by VarScan (using default parameters) were novel variants, and the majority of these novel variants were predicted to be damaging (Fig. 7 and Fig. S9). In contrast, only a minority of variant calls (0.4–9.7% for 1KG exome; 0.8–4.2% for 1KG targeted exon; 0.1–2.1% for SRP019719 exome samples) were known to be present at low frequency (<1%; see Methods) in the overall population, and this was true for all samples using all variant calling strategies. Importantly, this abnormally high proportion of novel variants matched the considerably decreased recovery rate between exome and targeted exome datasets, which we have been shown to be caused by a very low recovery rate among novel variants (Fig. 3, Tables S8 and S11).

Most GATK variant lists had a similar proportion of novel variants, except when the GATK UnifiedGenotyper was run without quality score recalibration in the 1000 Genomes samples (Fig. 7). However, trend doesn't apply to the SRP019719 samples (Fig. S9), emphasizing that there were other factors that can influence these results. The pre-processing steps had a modest impact on the VarScan frequencies, but the VarScan parameters had a very strong impact on the results. Namely, the frequency of novel variants was extremely high when running VarScan with default parameters, which corresponded to a decrease in the overall recovery rate (Fig. 3). However, the distribution for VarScan variants called with conservative parameters looked very similar to the GATK distributions, and these variants had a high recovery rate between the 1KG exome and targeted exon samples (Fig. 3).

ANNOVAR can also annotate variant frequencies for indels. However, small indels have not been characterised as well as SNPs (for example, there are considerably fewer indels in dbSNP (*Sherry et al., 2001*), compared to SNPs) and damaging predictions focus primarily on SNPs. Additionally, there are almost no indels in the coding regions of chromosome 20 for the 1000 Genomes targeted exon dataset. Nevertheless, Fig. S10 shows the variant frequencies for indel calls in the 1KG and SRP019719 exome datasets. Similar to the SNP distributions, VarScan-Cons contained the least number of novel indels (among the VarScan comparisons), which was likely associated with a lower false discovery rate. Also, novel indels were more common in GATK HaplotypeCaller variants than GATK UnifiedGenotyper variants. Arguably, this could indicate that the higher number of indels called by the HaplotypeCaller was also associated with a higher false discovery
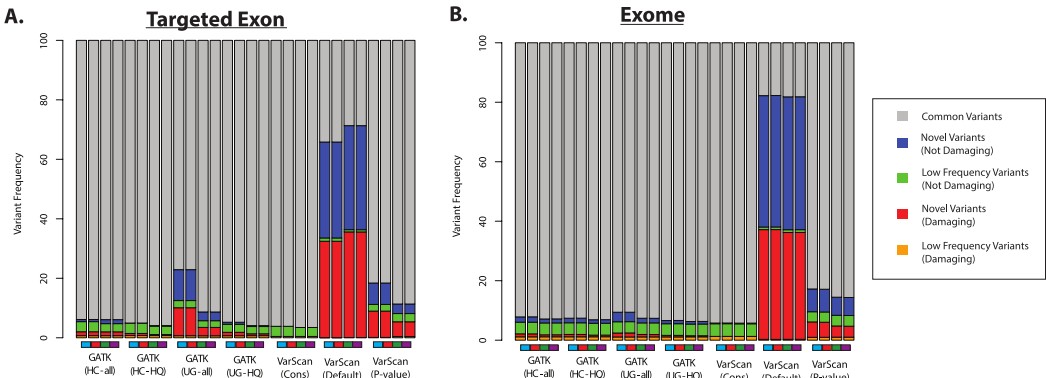

**Figure 7 Distribution of ANNOVAR annotations for coding SNP variants (1KG—chr20).** (A) Distribution of variant types defined in the ANNOVAR exome report for selected 1000 Genomes (1KG) targeted exon samples ($n = 14$). Variants are classified based upon population frequency and damaging prediction (see Methods). Low frequency variants (MAF $< 0.01$) that are displayed in orange if they are predicted to be damaging and are displayed in green if they are not predicted to be damaging. Novel variants are displayed in red if they are predicted to be damaging and are displayed in blue if they are not predicted to be damaging. Although all samples should contain some novel variants, a high proportion of novel variants are expected to correlate with a high false positive rate. Seven variant calling strategies were tested (GATK UnifiedGenotyper and HaplotypeCaller, with and without filtering low quality variants; VarScan with 3 sets of parameters, see Methods). "VarScan-Cons" is the most conservative set of parameters for VarScan. Each variant caller was also tested with 4 preprocessing conditions, corresponding to the colored boxes under the bar plot: variants called using both GATK indel realignment and quality score recalibration (purple), indel realignment only (red), quality score recalibration only (green), or neither (blue). (B) Same as (A), but for selected 1000 Genomes exome samples ($n = 12$, Table S2).

rate, but the variant frequencies alone are not sufficient to prove this to be the case because there may have been technical limitations in discovering the indels without using the GATK HaplotypeCaller. However, lower concordance for HaplotypeCaller indels between technical replicates (Fig. S2) supports the hypothesis that the HaplotypeCaller indels had a higher false discovery rate. Similarly, recovery of indels was higher for UnifiedGenotyper variants than HaplotypeCaller variants in the SRP019719 sample with both exome and amplicon sequencing data (Fig. S11). Interestingly, VarScan-Cons had the highest recovery rate of amplicon sequencing indels for that sample. Although the number of samples available for useful comparisons was limited, these results provide evidence from two independent cohorts that the GATK HaplotypeCaller indels are less reproducible than GATK UnifiedGenotyper indels.

## Overlap of high-quality GATK and VarScan variant calls

Given the previous comparisons, we believe the highest quality variant calls can be made by excluding GATK variants with LowQual flags (e.g., UG-HQ and HC-HQ) and using the conservative parameters defined in this manuscript when running VarScan (e.g., VarScan-Cons). Because the number of variants called increased when running the indel realignment and quality score recalibrator steps, we compared coding variant lists produced using variant callers with "Full Pipeline" pre-processing steps. In the 1000 Genomes samples, the variant caller overlap was quite high (Fig. 8 and Fig. S12).

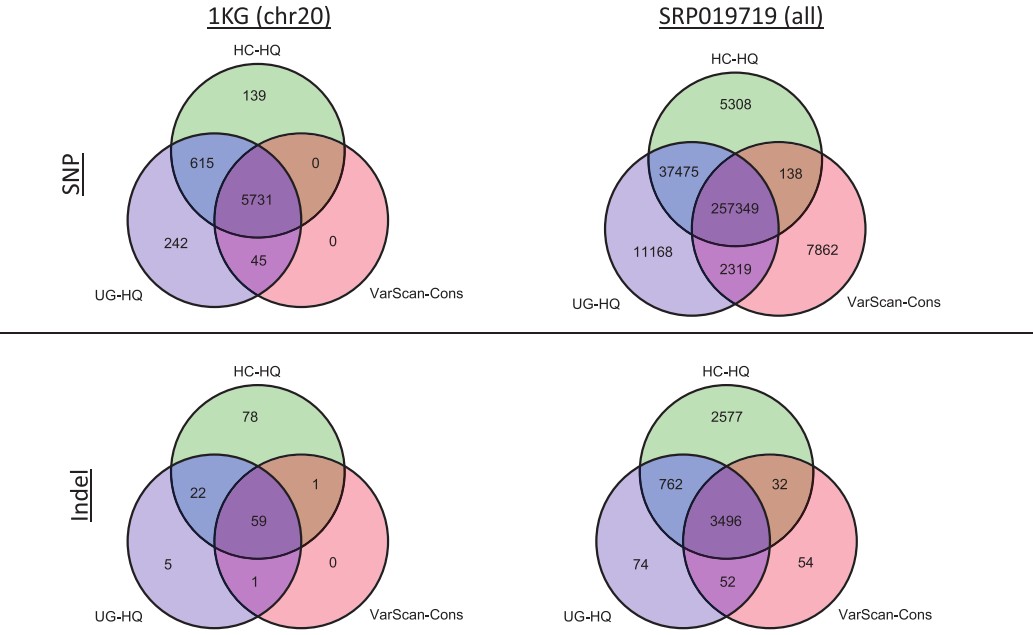

**Figure 8 Exome variant caller overlap (coding variants).** All coding variants were tabulated for all exome samples (1KG *n* = 12; SRP019719 *n* = 15). Only coding variants on chromosome 20 were considered for the 1000 Genomes (1KG) samples, but all coding variants were considered for the SRP019719 samples. In order to simplify presentation of these results, we focused on the highest quality variant calls for each variant calling strategy: GATK UnifiedGenotyper with low-quality variants removed (UG-HQ, blue), GATK HaplotypeCaller with low-quality variants removed (HC-HQ, green), and VarScan using a custom set of conservative parameters (VarScan-Cons, red). Similarly, only variants subject to GATK indel realignment and quality score recalibration ("Full Pipeline") are considered for this comparison. To show the different concordance rates, SNPs are presented at the figure and indels are presented at the bottom of the figure. Almost all VarScan-Cons variants were also called by GATK (either HaplotypeCaller or UnifiedGenotyper). All three variant callers called a similar number of SNPs, but GATK HaplotypeCaller called more indels than either GATK UnifiedGenotyper or VarScan-Cons.

For example, all SNPs called using VarScan-Cons were also called using the GATK HaplotypeCaller. Additionally, all indels called using VarScan-Cons were also called by either the GATK HaplotypeCaller or the GATK UnifiedGenotyper. Because only chromosome 20 variants can be reported for the 1000 Genomes dataset in this manuscript, it may also be useful to see the overlap for the genome-wide variant lists from SRP019719 (Fig. 8). This time, the VarScan-Cons variants were not completely recovered by using one or both GATK variant callers, but there was still only a minority of variants that were not represented in either list of GATK variants (2.9% of VarScan-Cons SNPs and 1.5% of VarScan-Cons indels). This emphasizes that VarScan-Cons calls robust variants that would almost always be called when using GATK.

The high recovery of VarScan-Cons variants may result from high specificity with decreased sensitivity, which is potentially the biggest drawback to this strategy. The degree to which VarScan-Cons recovers the set of high-quality GATK variants varied between datasets. Among the 1000 Genome variants on chromosome 20, 87% of UnifiedGenotyper SNPs and 88% of HaplotypeCaller SNPs were also called by VarScan-Cons in the exome samples. For the targeted exon samples, 62% of UnifiedGenotyper SNPs and 68% of HaplotypeCaller SNPs were called by VarScan-Cons in the targeted exon sample; similarly, 69% of UnifiedGenotyper indels but only 38% of HaplotypeCaller indels were also called by VarScan-Cons in the 1KG exome samples. The low indel overlap was likely due to the small number of coding indels on chromosome 20. Accordingly, the SRP019719 exome samples show similar statistics for SNP overlap (84% recovery for UnifiedGenotyper SNPs, 86% recovery for HaplotyperCaller SNPs) but much better results for UnifiedGenotyper indels (81% recovery). There was still only 51% recovery for HaplotypeCaller indels among VarScan-Cons indels, but it is possible that the indels uniquely called by GATK have a higher false discovery rate. An increased false discovery rate for GATK HaplotypeCaller indels has been previously reported (*Lescai et al., 2014*) and this conclusion is supported by evidence from this study (Figs. S2, S10 and S11).

### Run-times scale differently for GATK versus VarScan

Figure S13 shows run-times for the entire variant calling pipeline for the 1KG samples. The run-time for each step is reported in Tables S12–S14, with the average run-times for the variant calling step itself shown in Table S15 (for the "Full Pipeline" samples). Among the targeted exon samples ($n = 14$), the GATK UnifiedGenotyper had the longest run-time (Fig. S13A). However, the run-time for all samples was less than 5 h, so run-time was not a severe limiting factor for either VarScan or GATK (for targeted exon samples, compared to the exome samples). Adding a *p*-value filter did not significantly affect the run-time for VarScan; in fact, the run-time for VarScan was essentially the same regardless of what parameters are used for analysis. In contrast, the 1KG exome samples showed a wide range of run-times (Fig. S13B), with the HaplotypeCaller having a very noticeably longer run-time compared to the other pipelines. Quality score recalibration also caused a noticeable increase in run-time. As expected, the newest version of GATK (with or without read reduction) decreased the run-time for GATK HaplotypeCaller (Tables S15–S17). However, the run-time was always shorter for VarScan compared to GATK.

For the SRP019719 exome samples, the most obvious trend was the run-time for the GATK HaplotypeCaller was considerably longer when running base recalibration without prior indel realignment (Fig. S14). Likewise, population frequencies among variants (Fig. S9) indicate that the variants called using base recalibration alone were probably not reliable (for these particular samples), but the variants called using the full pipeline (indel realignment + base recalibration) were more likely to have a lower false discovery rate (similar to the "No Preprocessing" and "Realign Only" results). Two factors mostly likely caused this increased-run time are (1) quality scores were on a different scale than the 1KG samples and (2) the servers running the variant callers had considerable concurrent usage

(where variability in usage may explain why the run-times for some samples were much worse than other samples). More specifically, the quality scores were identified as miscoded by GATK, and it appears that the quality scores could not be successfully fixed at the base quality recalibration step (see Texts S1 and S2 for more details). In contrast, the miscoded quality scores could be successfully shifted by GATK prior to quality score recalibration or at the variant calling step (if neither indel realignment nor quality score recalibration was run). Therefore, we do not expect this trend to apply to most samples that are processed in accordance to GATK "best practices".

Run-times can also be affected a running joint variant caller instead of running variant callers separately for each sample. One potential advantage to this strategy is that it can increase sensitivity in cohorts of related individuals (*Nielsen et al., 2011*). One disadvantage is that joint variant calling can have substantial memory requirements for large cohorts. However, joint variant calling could be successfully run on all the groups of samples in this study, with a substantial decrease in run times for the GATK joint variant caller (Table S16). In contrast, joint variant calling considerably increased the runtime for VarScan, but the total run-time for single-sample VarScan calls was still less than the run-time for the joint variant calls in GATK (Table S16).

## Comparison to other variant calling benchmarks

In order to provide a way to quickly compare VarScan-Cons results to other variant callers (with other pre-processing pipelines), the Genome Comparison & Analytic Testing (GCAT) website was used to compare six sets of variants calls for a standard sample that has been characterized by the Genome in a Bottle Consortium (*Zook et al., 2014*). This sample contained 100 base pair Illumina reads for a 150× Exome sample (making most comparable to the samples in this study). This is not the only available tool that can benchmark variant caller algorithms (*Nevado & Perez-Enciso, 2014*; *Talwalkar et al., 2014*), but it was selected based upon the ability to compare a targeted sequencing sample without the use of simulated data (similar to the design of the current study). Links to reports for each variant list are provided in Table S18. Although the presence of a single (albeit well-characterized) sample is a limitation to the strategy, the GCAT report shows results that complement and enhance this study. For example, VarScan-Cons showed a decrease in sensitivity when recovering Omni SNP array genotypes but also showed the highest specificity and precision rate, when compared to variant calls with no pre-processing steps as well as with the full pre-processing pipeline (Fig. S15). The GCAT reports show variant counts and overlap of variants between variant callers that match what is expected from this study (Fig. S16), although it is important to note that these are genome-wide counts (similar to Fig. 6 and Fig. S8) whereas most statistics are providing in this study are for coding variants (such as Figs. 8 and 9). In other words, VarScan-Cons shows similar on-target variant calls to the high-quality GATK variant lists, but GATK made more off-target calls in lower coverage regions. Importantly, the GCAT report also showed the transition to transversion (Ti/Tv) ratio for SNPs for novel as well as common SNPs. The average value for Ti/Tv is between 2.1 and 2.8, so lower ratios can be an indication

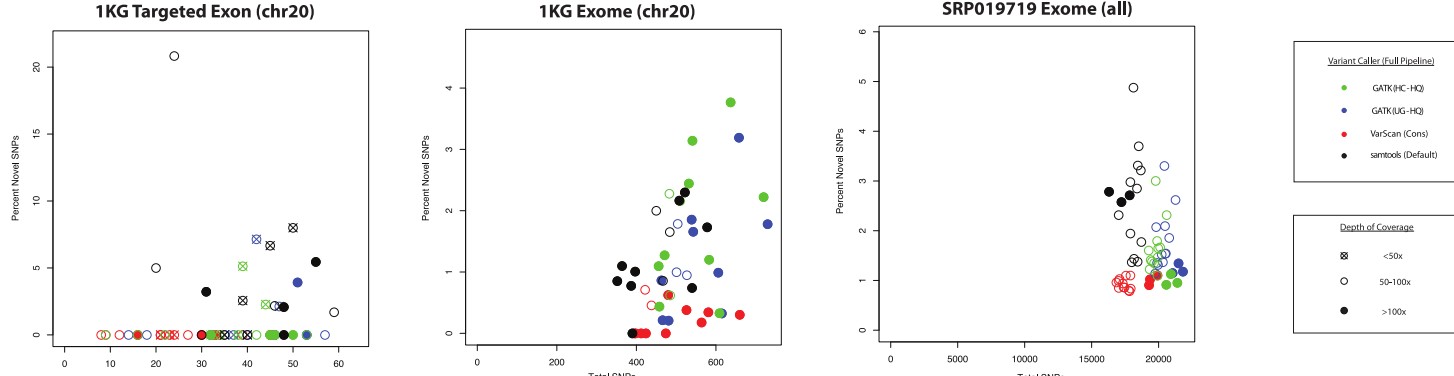

**Figure 9** **QC metrics to estimate specificity versus sensitivity for variant callers (coding SNPs).** For each of the three datasets characterized in this study (1KG targeted exon, $n = 14$; 1KG exome, $n = 12$; SRP019719 exome, $n = 15$), the number of coding SNPs called per sample is plotted along the *x*-axis and the proportion of novel variants is plotted on the *y*-axis. In order to simplify presentation of these results, we focused on the highest quality variant calls for each variant calling strategy: GATK UnifiedGenotyper with low-quality variants removed (UG-HQ, blue), GATK HaplotypeCaller with low-quality variants removed (HC-HQ, green), and VarScan using a custom set of conservative parameters (VarScan-Cons, red). Additionally, an unfiltered set of variants called via samtools are plotted in black. Only variants subject to GATK indel realignment and quality score recalibration ("Full Pipeline") are considered for this comparison. The shape of the data point corresponds to the depth of on-target coverage: <50x coverage is represented as an X in an open-circle, 50–100x is represented as an open circle, and >100x is represented as a filled circle. If the novel percentage was tightly correlated with the actual false positive rate and the number of variants was tightly correlated with the actual sensitivity of the variant caller, than the ideal variant caller would show a cluster of data points in the bottom-right hand corner of the plot.

of an increased false discovery rate. As we would expect from the reproducibility results from this experiment, novel VarScan-Cons variants showed a considerably higher Ti/Tv ratio (and therefore considerably lower false discovery rate) that was more similar to the expected value than the novel high-quality GATK variants (Fig. S17).

## Application of QC metrics to other variant callers

Although we were primarily interested in comparing VarScan (with various sets of parameters) to GATK, we also examined similar metrics for other variant callers. Given that the distribution of rare and novel variant frequencies was similar for the 1000 Genomes targeted exon and exome datasets, we used the targeted exon samples to test additional variant callers. More specifically, we tested freebayes (*DePristo et al., 2011*) and samtools (*Li et al., 2009*), using the "No Preprocessing" and "Full Pipeline" alignments. It is clear that freebayes yields too many novel variants (Fig. S18), so it was not tested on any further datasets. Of course, changes in parameters and/or downstream filtering of variants may result in a decrease in the frequency of novel variants; however, these two lists of variants (called with default parameters) are meant to serve as examples of how the quality control metrics in this dataset could be applied more broadly. Interestingly, the pre-processing steps appeared to have minimal effect on the 1000 Genomes targeted exon and exome datasets, but running the "Full Pipeline" for pre-processing considerably decreased the proportion of novel variants for samtools variant calls in the SRP019719 exome dataset (Fig. S18). Most samtools variants overlapped VarScan-Cons variants as well as high-quality GATK UnifiedGenotyper and HaplotypeCaller variants (Fig. S19).

The ANNOVAR frequency plots (like those shown in Fig. 7, Figs. S9 and S18) are a useful metric, but they only show average trends across the dataset and do not show how the frequency of novel variants compares to the total number of variants called. If the number of conditions is reduced to only those using the "Full Pipeline" for pre-processing, then one can visualize the comparison between variant callers (Fig. 9). There was more variability in the proportion of novel variants for the targeted exon dataset (compared to either exome dataset), but this may relate to the lower number of total SNPs and/or the lower on-target coverage (Tables S1–S3). While it is not necessarily safe to assume that the false discovery rate substantially varies when the proportion of novel variants is between 0 and 6% (which is the observed range for these variant callers in the exome datasets), the proportion of novel variants is universally low for all VarScan-Cons calls in all three datasets while there is more variability in the proportion of novel variants in the GATK and samtools variant calls. VarScan-Cons and the high-quality GATK variant calls showed similar distributions of supporting reads (Fig. 5), so the larger number of novel variants was not being driven by variants that were extremely likely to be false positives (Fig. 4). Nevertheless, there was a decrease in recovery rate (9–20% for 5–11 UnifiedGenotyper SNPs and 30–50% for 1–3 HaplotypeCaller SNPs) for a limited number of novel, high-quality GATK variants compared to common, high-quality GATK variants (Table S8, Fig. 3), but the VarScan-Cons criteria were so strict that no novel variants were called within targeted exon regions on chromosome 20. Although we cannot make a definitive conclusion using the data in this current study, the difference in the percentage of novel SNPs could be explained if VarScan-Cons had a relatively lower false discovery rate among novel, high-quality variants. This hypothesis can be supported by the observation that novel VarScan-Cons variants showed a substantially higher Ti/Tv ratio that is much more similar to common variants (Fig. S17), so it is plausible that novel VarScan-Cons variants indeed have a decreased false discovery rate relative to novel GATK variants for array-based targeted sequencing experiments.

## DISCUSSION

Although running VarScan with default parameters (with the functions defined in the Methods section) was shown to result in an unacceptably high false discovery rate (Fig. 3), running VarScan with a conservative set of non-standard parameters (referred to as VarScan-Cons in this study) can produce a reliable set of variants with an overall concordance between sequencing technologies that was at least as strong as high-quality GATK variants (Figs. 1–3, Figs. S1–S3, S11 and S17). Almost all variants called with VarScan-Cons were also called using the GATK HaplotypeCaller or GATK UnifiedGenotyper, with a modest decrease in sensitivity for SNPs (Figs. 1–2, Fig. S15). However, given that the high-quality GATK variants have a similar overall specificity to VarScan-Cons (Figs. 1–3, 7 and Fig. S15), increased specificity is one advantage to using GATK HaplotypeCaller or GATK UnifiedGenotyper (Fig. 6 and Fig. S16). Although less obvious than the VarScan variant lists, the high-quality GATK variant lists consistently show improved recovery rate for targeted exon variants (Fig. 3) and lower percentages

of novel variants (Fig. 7), even without additional pre-processing steps. So, we would recommend VarScan users use the VarScan-Cons parameters and GATK users to filter out variants flagged as low-quality. In all three cases (VarScan-Cons, high-quality GATK HaplotypeCaller, high-quality GATK UnifiedGenotyper), indel realignment and quality score recalibration affected only a minority of variants (Figs. S6–S7), so users could arguably skip those pre-processing steps and still get similar results (if a modest decrease in overall validation rate was acceptable). GATK HaplotypeCaller called a substantial number of indels not called using VarScan-Cons (as well as GATK UnifiedGenotyper), even after removing variants that were flagged as low quality. However, it is possible that the GATK HaplotypeCaller indels also have an increased false discovery rate (Figs. S2, S10 and S11, as reported in *Lescai et al. (2014)*, so it will be important to see if this is independently validated in other studies.

It is important to note that most of the statistics used for calling VarScan variants can be extracted from the .vcf file containing the GATK variants. In other words, users may not need to run both GATK and VarScan. For example, the distributions of supporting read frequencies look very similar for the high-quality GATK variants and the VarScan-Cons variants (Fig. 5), so excluding GATK variants flagged as low-quality can already produce results that are more similar to VarScan-Cons. This study presents some limited evidence that VarScan-Cons variants may have a lower false discovery rate among novel variants in array-based enrichment designs, compared to the high-quality GATK variant calls (Table S8 and Fig. S17). So, GATK users may be able to improve the accuracy of their novel variant calls by imposing stricter requirements on frequency of supporting reads, number of supporting reads, and/or read depth. This is important because high-throughput sequencing studies often emphasize novel, rare variants as interesting candidates (*Worthey, 2013*). Based upon the results of this study, we expect that variant calling comparisons that only focus on complete sets of variant calls (where most variants are likely to be common variants) may give false confidence in the accuracy of variant calls for novel variants. However, additional evidence is necessary to confirm that VarScan-Cons defines more accurate novel SNPs than the high-quality GATK variants. Nevertheless, we believe that this study provides good evidence that users should be suspicious of lists of variants with substantially increased novel variant frequencies, such as those with novel variant frequencies of 50% or higher (like VarScan-Default).

This study focused on variant calling in (most likely) normal human samples due to the availability of a large amount of public validation data. However, the strategies described in this study may not apply equally well in all circumstances. For example, sometimes variants may be present in a minority of cells in a sample (such as a heterogeneous tumor), and it may not be safe to make assumptions about the ploidy of the sample (which might affect the usefulness of the GATK HaplotypeCaller, for example). In fact, the results of this study emphasize the need to have specialized experimental protocols (such as circle sequencing (*Lou et al., 2013*), Duplex-Seq (*Schmitt et al., 2012*), etc.) for calling such variants because variants with low numbers of supporting reads show an extreme decrease in recovery rate between different targeted sequencing designs (Fig. 4 and Fig. S4, and

Tables S8 and S11). Likewise, somatic variant calling is also an area of interest (*Roberts et al., 2013*; *Wang et al., 2013*; *Xu et al., 2014*) that would typically utilize different variant calling tools. Additionally, the indel metrics in this study only apply to small indels: large indels and structural variants require a different set of algorithms. Nevertheless, we think the strategies described in this study (comparing the proportion of novel variants, using filters for high-quality variants, etc.) can be useful to help other scientists prioritize variant calling strategies for their own data.

## ACKNOWLEDGEMENTS

We would like to thank Laura Clarke for providing technical assistance with the 1000 Genomes data, Jason O'Rawe and Gholson Lyon for answering questions about the (*O'Rawe et al., 2013*) exome dataset and providing access to the Human610 SNP chip data and providing helpful comments to improve this manuscript, and Illumina technical assistance for answering questions about the Illumina SNP chips. We would also like to thank Han Fang and an anonymous reviewer for helpful comments.

### Funding

This study is supported by National Institutes of Health (Comprehensive Cancer Center Grant P30 CA33572) and City of Hope National Medical Center institutional funding. The funders had no role in study design, data collection and analysis, decision to publish, or preparation of the manuscript.

### Grant Disclosures

The following grant information was disclosed by the authors:
National Institutes of Health: P30 CA33572.
City of Hope National Medical Center.

### Competing Interests

The authors declare there are no competing interests. Aaron W. Adamson, Susan L. Neuhausen, and Xiwei Wu are employees at the City of Hope National Medical Center.

### Author Contributions

- Charles D. Warden conceived and designed the experiments, performed the experiments, analyzed the data, wrote the paper, prepared figures and/or tables, reviewed drafts of the paper.
- Aaron W. Adamson, Susan L. Neuhausen and Xiwei Wu conceived and designed the experiments, reviewed drafts of the paper.

### Supplemental Information

Supplemental information for this article can be found online at http://dx.doi.org/10.7717/peerj.600#supplemental-information.

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
