# Peer review of "Detailed comparison of two popular variant calling packages for exome and targeted exon studies"

_PeerJ, doi:10.7717/peerj.600_

## Round 0.1 · original submission · Major Revisions

· Academic Editor

Major Revisions

Dear Charles

Please consider all the comments by the independent reviewers. Both have requested additional comparisons and clarification of findings. These changes are considered essential to develop your initial manuscript into a more substantial and independent body of work of use to the wider community.

·

Basic reporting

The reviewer for this paper is Gholson Lyon. He claims responsibility for the below suggestions, and he did seek input from his colleagues, Han Fang and Jason O’Rawe, who work with him at Cold Spring Harbor Laboratory.

I would recommend the authors review this section regarding PeerJ papers, as our below suggestions relate to this:
• The article should include sufficient introduction and background to demonstrate how the work fits into the broader field of knowledge. Relevant prior literature should be appropriately referenced.
• The submission should be ‘self-contained,’ should represent an appropriate ‘unit of publication’, and should include all results relevant to the hypothesis. Coherent bodies of work should not be inappropriately subdivided merely to increase publication count.

1. On a fundamental and conceptual level, I do not believe that what the authors have submitted here represents a “self-contained” body of work. I detail some possible additions and suggestions below, which I hope are helpful, that might improve this work toward being acceptable for scholarly publication.

General comments/suggestions to improve clarity

2. Although I tend to agree that the sentiment of the field seems to be that GATK is a well-performing analytical software tool set, this below sentence is not entirely accurate.
“Multiple studies have previously compared variant callers for short-read sequencing data, which have often indicated that the variant callers available in the Genome Analysis ToolKit show the best performance”.

In particular, this (accepted for publication) manuscript shows improvements over GATK, specifically in INDEL detection: Accurate detection of de novo and transmitted INDELs within exome-capture data using micro-assembly (http://biorxiv.org/content/early/2014/06/18/001370).

Also, the GCAT website might be helpful to the authors, as this website details various comparisons that other users have performed: http://www.bioplanet.com/gcat

3. In the abstract, the statement “More specifically, we compared four lists of variants called by GATK (using the UnifiedGenotyper and the HaplotypeCaller algorithms, with and without filtering low quality variants) and three lists of variants called using VarScan (with varying sets of parameters)” is somewhat hard to interpret on its own, without reading the rest of the manuscript. More precise wording could improve this sentence.

4. In the abstract “in a high quality gene list, with high concordance (>97%) when compared to high quality variants called by the GATK UnifiedGenotyper and HaplotypeCaller” the words ‘gene’ and ‘variants’ seem to be used interchangeably, so it is difficult to understand what was being compared. The clarity of this statement could be improved if the authors try to re-word, or briefly expand, this statement.

5. On line 48, the term/statement “proportion of unknown frequency variants” is introduced without a clear definition. This term is somewhat difficult to intuitively understand. It might improve clarity to either define this term as soon as it is introduced or use another term like “proportion of novel variants” or “proportion of previously un-described variants” or something that is more intuitively understood.

Experimental design

I would recommend the authors review this section regarding PeerJ papers, as our below suggestions relate to this:
• The submission should clearly define the research question, which must be relevant and meaningful.
• The investigation must have been conducted rigorously and to a high technical standard.
• Methods should be described with sufficient information to be reproducible by another investigator.

1. Fundamentally, I am not sure what exactly the authors’ relevant and meaningful research question is. The authors could re-write the intro and results to try to make this clearer.

2. MiSeq validation data for the sample K8101-49685s was deposited in the Sequence Read Archive (http://www.ncbi.nlm.nih.gov/sra) under project accession number SRX386284. It could be informative to employ the use of these data. Because one can only measure the concordance between a variant caller and a SNP chip call set across some common sites, the authors could also try to validate the INDEL call set, and rare/unique SNPs, using these data.

3. Since GATK v3.1.1 supports multi-sample calling, it might be helpful to report the run-time difference if one uses this functionality. It would be interesting to see if pooling the samples and calling variants simultaneously would increase/decrease run time.

4. Figure 2a seems to represent the number of SNP rather than the run time analysis.

5. When first introducing a pipeline in this manuscript, it might benefit the reader to know the pipeline version number and to make clear, if necessary, that the same version was used throughout. What are the versions of BWA, Picard and ANNOVAR used in this study? Although the authors, for the most part, detail the pipeline parameterizations throughout the document, it might also be useful to provide an ancillary document with pipeline commands so that these are all in a single location for readers to refer to.

6. In the methods section on line 69 “and validated variants come from ALL.chr20.exome_consensus_validation_454.20120118.snp.exome.sites.vcf.gz”, what method(s) were used to validate these variants?

Validity of the findings

I would recommend the authors review this section regarding PeerJ papers, as my below suggestions relate to this:
• The data should be robust, statistically sound, and controlled.
• The data on which the conclusions are based must be provided or made available in an acceptable discipline-specific repository.
• The conclusions should be appropriately stated, should be connected to the original question investigated, and should be limited to those supported by the results.
• Speculation is welcomed, but should be identified as such.

1. The authors did not report run time improvements for the samples tested on the authors servers. Did the authors take full advantage of the optimizations in GATK v3.1.1 (e.g., -pairHMM VECTOR_LOGLESS_CACHING)? It should be noted that the BQSR step is not yet optimized to decrease run time in this version.
See here: http://gatkforums.broadinstitute.org/discussion/3930/version-highlights-for-gatk-version-3-1

2. I am not entirely convinced that the methods used by the authors to estimate accuracy are valid. On line 199 and 200, the statement "indicate that the variants called using base recalibration alone were probably not reliable" seems like a judgment call made without sufficient support from the data presented within this text. On line 209, "we assume that the total number of variants is correlated with sensitivity", this assumption needs the appropriate justification, perhaps by means of generating additional validation data demonstrating the validity of this assumption or with an external citation. Otherwise, the text should be updated to reflect that this assumption may or may not be reasonable. On line 288, “we assume that most variants should have known frequencies, and that an overrepresentation of variants of unknown frequencies should correspond to an increased false positive rate”, again, this assumption needs to be based on validation data or on previous work to be considered a reasonable one, otherwise the text must reflect its uncertain nature. It may be a reasonable assumption, but it may not be.

3. On line 243 "corroborate the conclusion that there was a technical problem in producing these variant calls", this needs to be explained in more detail. It is not clear what the authors mean by a ‘technical problem’.

4. On line 397 "a reliable set of variants that should show a high validation rate", I am not convinced that this has been sufficiently supported by the evidence in this manuscript. There needs to be a better quantification of the true pipeline error rates by means of a validation experiment, or by demonstrating, through some other means, that the assumptions that have been made to arrive at this conclusion are valid.

Comments for the author

1. One general suggestion is that the authors could perform additional validation experiments to confirm and/or support some of the assumptions.

Reviewer 2 ·

Basic reporting

The authors conducted a detailed comparison between GATK and VarScan with three sets of parameters, and showed that VarScan can generate reliable variants with a conservative parameter set. They also evaluated the effects of local realignment against indels and per-base quality score recalibration to the variant sets. In addition to the concordance with validated SNPs or SNPchip data, the annotation features of variants, including frequency and functional prediction, were used as measures for accuracy of variant sets.

Experimental design

The authors carefully selected publically available data for this study. 14 targeted exon (for 1000 genes) samples and 12 exome samples from the same healthy individuals in the 1000 Genomes project (1KG), and 15 Illumina exome samples from SRP019719. The individuals from 1KG also have validated SNPs, and Omni SNP chip data. Functional annotation by ANNOVAR was included. Besides the rediscovery rate, percentages of novel variants and damaging variants were introduced as additional measures for quality control. Running time was compared. And comparison with SAMtools and FreeBayes was touched.

Validity of the findings

The metrics were defined clearly, and interpreted appropriately.

Comments for the author

1. Using the percentages of novel/damaging variants as measures brings novelty to this work. This should be emphasized.
2. As the manuscript shows, VarScan with a conservative parameter set generated reliable variant sets, but lost about 10% in sensitivity. Then what is the value to use VarScan rather than GATK, given the running time of GATK (UC/HC) is endurable? Is VarScan more suitable for special cases, like cancer? Arguments or discussions are in need. Also GATK’s strengths should be addressed concisely.
3. As the authors pointed out, the recovery of validated SNPs was not robust. Therefore it was not surprising that no effects of the preprocessing procedures were observed for most settings. However, it is surprising to see that the effects were again unobservable for the recovery of SNP chip data (shown in Figure 6B). Please change the scale to show the differences.
4. As the targeted exon samples are from the same 1KG individuals, a piece of work checking the consistency between the two sets will be helpful.
5. The term “minimum p-value threshold” for VarScan needs to be changed to “p-value threshold”, and please confirm that 0.99 is the default cutoff.
6. “CentOS Red Hat 4.1” in line 151 is confusing. Is not CentOS different from Red Hat?

---

## Round 0.2 · Minor Revisions

· Academic Editor

Minor Revisions

As you will see the reviewers comments suggest changes to aid clarity and add depth to the paper. I think these are worthy of addressing in a revised draft. I hope that you are able to make these changes relatively swiftly. I will oversee these changes and It is expected that no further review will be required, so your paper should be accepted and published will little delay. Many Thanks for your interest and contribution to PeerJ. I look forward to the revised version.


Richard

·

Basic reporting

This paper was re-reviewed here by Gholson Lyon and Han Fang.

The authors have done a good job of responding to reviewer comments, and this paper should be of interest to anyone using VarScan. We list some points for the authors to consider below, which could serve to clarify the paper further. We offer these as suggestions only, which we hope the authors might address to further improve their paper. We do not feel that we need to re-review this paper for these minor revisions.

1. Line 16-18:
This statement is not entirely accurate. GATK itself serves as a tool kit other than just a variant caller. Can you clarify when citing different publications, in which the comparisons were performed with HaplotypeCaller or UnifiedGenotyper?

2. Line 54:
It might be a good idea to define “novel” the first time it appears in the paper or describe this with details here. In addition, the online databases are constantly being updated, especially for dbSNP. Please specify the version and data of database when you say a variant is “novel”, because a “novel” variant might no longer be “novel” if it is picked up by a database with a newer release.


3. Line 59:
You mentioned false positive rate (FPR) here. But is it possible that you might be talking about false discovery rate (FDR) instead because FDR= FP/(FP+TP) while FPR=FP/(FP+TN). Please define what you mean by false positive rate in the methods or use false discovery rate.

4: Line122-127:
Descriptions and definitions about concordance are still very confusing. Please consider using recovery rate and concordance rate separately to measure different scenarios. Also, the following way of calculating concordance for technical replicates seems hard to interpret (2 * Number of Overlapping Variants) / (Sum of Variants from both samples). Please consider using (Intersection of A and B)/(Union of A and B) as this metric is commonly used in different studies.

5. Line186-203:
The assumption of treating these two samples as technical replicates seems problematic. Because the following questions are unclear for these four data: SRR013654, SRR013709, SRR017908, SRR018122.
1. What was sequencing platform? What was the sequencing center?
2. What library preparations protocols were used?
3. What are read lengths?
4. What is the mean coverage?
Until we control for the above effects, the statements in this section are hard to interpret because you might be going after batch/sequencing effects, instead of using technical replicates to understand algorithmic differences. One good example is the much higher discrepancy you saw in the sample NA18510. We would recommend further clarification in this section.

6. Line214-228:
Here you jump from concordance to recovery rate several time, which is quite confusing for the readers. Please clearly define them in the methods.

7. Line256-298
In the previous study, the SRP019719 data were used to validate pre-selected variants so the amplicons were designed to place the variant in the middle. As you could observe from this section, PCR amplifications might introduce “errors” into this kind of data, where a caller might possibly call some likely false positive variants either near the edge of the amplicon or with insufficient percentage of reads supporting the allele. Thus, in order to control for such effects, it is very important for one to be more stringent and require a larger percentage of reads supporting a variant with extremely high coverage data. I think the authors were trying to deliver this message but the current writing might confuse common readers. Please clarify this in a more concise way in this section in order to make it more readable.

8. Axis labels of many figures are too small.

Experimental design

no other comments

Validity of the findings

no other comments

Comments for the author

Thank you for your detailed response to our reviews.

Reviewer 2 ·

Basic reporting

The authors conducted a comprehensive comparison between GATK and VarScan, and evaluated the effects of local realignment against indels and per-base quality score recalibration to the variant sets. They showed that VarScan can generate reliable variants with a conservative parameter set, with highest specificity demonstrated by the Genome Comparison & Analytic Testing standard sample. And they also showed that the novel variants had higher false positive rate mainly due to insufficient support reads. This revised version is more sound, and better organized than the original one.

Experimental design

The authors carefully selected publically available data for this study. 14 targeted exon (for 1000 genes) samples and 12 exome samples from the same healthy individuals in the 1000 Genomes project (1KG), 15 Illumina exome samples from SRP019719 (one with MiSeq amplicon sequencing data for some variants), and the standard sample of Genome Comparison & Analytic Testing (GCAT). The individuals from 1KG also have validated SNPs, and Omni SNP chip data. The additional data enhance the study. Functional annotation by ANNOVAR was included.

Besides the rediscovery rate, proportions of novel variants and damaging variants were introduced as additional measures for quality control. Good efforts were devoted to novel variants, for which low supporting read frequency was shown as a cause of high false positive rate. Running time was compared. And comparison with SAMtools and FreeBayes was touched.

Validity of the findings

The metrics were defined clearly, and interpreted appropriately.

Comments for the author

English needs to be polished. Below are some suggestions (not complete):
1. In abstract, 'More specifically, we compare..., we compare..., and we compare...' is too intensive.
2. The tense should be unified into past in the results section, e.g. 'is' and 'was' mixed now.
3. Line 85, 'via samtools' -> 'using samtools', etc.
4. Line 191, 'are considered' appears twice, replaced by 'were considered'.
5. Delete line 193 'when using a more focused set of SNPs'.
6. Line 196, 'worse concordance' -> 'lower concordance'.
7. Line 197 'percent duplicates' -> 'percent of duplicates'.
8. Line 199 'than the NA18510 samples' -> 'than for the NA18510 samples'.
9. Line 258, 'the amplicon dataset more novel', a verb missing.
10 Line 501, 'strong a' -> 'strong as'.

---

## Round 0.3 · accepted · Accept

· Academic Editor

Accept

Thank you for the updated manuscript. I have passed a single typo to the editorial staff (line 151 clearity should read clarity). This should be picked up in your proof version, but please check.